# A Single-Day Training for Managers Reduces Cognitive Stigma Regarding Mental Health Problems: A Randomized Trial

**DOI:** 10.3390/ijerph19074139

**Published:** 2022-03-31

**Authors:** Michael Gast, Janina Lehmann, Elena Schwarz, Christian Hirning, Michael Hoelzer, Harald Guendel, Elisabeth Maria Balint

**Affiliations:** 1Department of Psychosomatic Medicine and Psychotherapy, University Medical Center, 89081 Ulm, Germany; michael.gast@uni-ulm.de (M.G.); janina.lehmann@uni-ulm.de (J.L.); elena.schwarz@uni-ulm.de (E.S.); christian.hirning@uni-ulm.de (C.H.); elisabeth.balint.research@gmail.com (E.M.B.); 2Sonnenberg Klinik gGmbH, 70597 Stuttgart, Germany; hoelzer.michael@sonnenbergklinik.de; 3Privatklinik Meiringen, 3860 Meiringen, Switzerland

**Keywords:** SMI, workplace intervention, stigma, stress management, managers

## Abstract

Background: Mental illnesses have received increasing attention in the work context in recent years, yet they are still often accompanied by stigma. One starting point for stigma reduction is interventions in the workplace. The present study evaluated a one-day workshop for managers in a large company. Method: Enrolled managers (*n* = 70) were randomly assigned to the intervention group and the waiting control group. The training included a theoretical section on mental and stress-related diseases as well as the interplay between work and health, group work on personal stress experience, theoretical input on dealing with mentally ill employees, and a group discussion on this topic along with case studies. Both groups completed the following questionnaires at baseline and three months after training: Effort–Reward Imbalance Questionnaire, Patient Health Questionnaire, Mental Health Knowledge Schedule, Social Distance Scale, and the Irritation Scale. Results: Compared to the waiting group, the intervention group showed a significant improvement in the Mental Health Knowledge Schedule (U = 417.00, *p* = 0.040) and an increase in the Irritation Scale (U = 371.50 *p* = 0.011). All other scales remained unchanged. Conclusion: The content and duration of the training were adequate to reduce cognitive stigma towards mental illness. However, the present approach was not sufficient for an improvement in the subjective stress level of the participating managers.

## 1. Introduction

Mental health disorders are one of the main causes of sickness absence and long-term disability in the modern workplace [1,2]. Compared to other diseases, the economic costs of mental health disorders are high. In Germany, the costs amounted to 121.11 billion euros in 2010 [3]. The majority of these costs are caused by common mental health disorders, primarily depression and anxiety disorders [4]. This corresponds to a high prevalence of mental illness in the population. For instance, the survey by Jacobi and colleagues found a 12-month prevalence of 9.3% for affective disorders, 15.3% for anxiety disorders, and 27.7% for having any mental illness [5]. 

One of the reasons for the high costs is the high rate of early retirement due to mental illnesses [6]. Another reason is the long duration of illnesses; the associated absence from work of 34 days at mean is higher than other common causes of sick leave [6]. In addition to these direct costs, there are indirect costs as well. These arise mainly when employees show up for work despite suffering from mental health problems [7]. This so-called presentism may come with reduced workplace performance and productivity as demonstrated in many investigations [8,9,10,11]. The economic costs associated with presentism even exceed those of absenteeism [12] and many employers have to deal with both cost types.

While mental illnesses of employees may have a high impact on companies, the conditions at the workplace themselves can also favor the development of disease. Examples of potentially risky conditions include, but are not limited to, long working hours, interruptions, time pressure, and interpersonal difficulties [13]. As there are many different factors influencing the development of stress-related symptoms, there are also many different theoretical approaches. One well-evaluated model is the effort-reward imbalance model. Effort-reward imbalance (ERI) describes an insufficient reward in relation to the invested effort [14]. This widely validated construct is related to increased risk of mental health disorders as well as cardiovascular disease, impaired employee well-being, and emotional exhaustion [15,16,17].

Managers are important factors within a company. They have a direct influence on the working conditions of their employees and thus influence their reward within the ERI model. Furthermore, they influence the stress of their employees through their leadership style or by acting as role models [18]. This interaction with employees is influenced by the stress of the manager. There is a correlation between stress and negative behavior towards employees: stressed managers show more negative behavior towards employees [19]. Leadership stress is also linked to less ethical decision-making and a reduced propensity to take a team perspective [20,21]. According to the Center for Creative Leadership, 88% of leaders say that work is the main source of their stress [22]. One source of stress is feeling unprepared for work demands. In a Canadian survey of managers, 45% of respondents reported that they had not received any training on mental illness or how to deal with employees with mental health problems [22]. Yet this may be urgently needed, as superiors are frequently the preferred contact person when it comes to communicating a mental illness [23]. Therefore it should be possible to reduce the stress of managers by training them to deal with stressed employees so that they feel better equipped to deal with the demands of their task.

A basic prerequisite for this is a non-stigmatizing attitude towards people with mental health problems. In general, stigma still represents a major issue in public health. Although improvement in overcoming job impairments by treating depression has been shown [24], individuals suffering from mental health problems often refrain from seeking treatment due to fear of stigmatization [25]. Despite many prosocial reactions, there is a tendency to increase social distance from mentally ill people [26]. Thus, one possible consequence of disclosing mental illness in the workplace is that it can negatively impact relationships with co-workers [23]. Common preconceptions in the workplace include the notion that mentally ill people are unpredictable and possibly violent as well as lacking the competence to meet task requirements [27]. These prejudices are also reflected in employers’ concerns about hiring people with mental health problems [28]. In addition to actual stigma, a further problem is the overestimation of expected stigma by people with mental illness [29]. For example, 55.6% of people with depression reported that they expected to be shunned by others because of their illness, but only 26.4% experienced rejection [30]. Another concern is potential negative consequences for the individual’s career after disclosure [31]. Stigma is one of the most important barriers to seeking treatment [32] and predicts avoidant coping [33]. Despite an improvement in mental health literacy in the general population, a meta-analysis by Schomerus and colleagues found in 2012 that stigmatizing attitudes have not decreased [34]. Therefore, actions to reduce stigma are a major challenge and an important task. 

Possible approaches to reducing stigma include direct contact with those affected, and dissemination of knowledge. Both approaches are suitable for reducing stigma [35]. Programs can be aimed at the general population or specific target groups [36]. As stigma may play an important role in working life, the workplace provides opportunities for interventions [37]. Regarding specific groups, the effectiveness of workplace interventions has already been demonstrated for military personnel, teachers, and police officers [38,39,40]. Due to the nature of their their role, stigma reduction interventions for company leaders have also been investigated [41,42,43]. A meta-analysis from Gayed and colleagues demonstrated their effectiveness in improving mental health knowledge, non-stigmatizing attitudes, and positive behavior towards affected employees [44]. A study conducted within a large Australian fire and rescue service demonstrated that programs designed to train managers on how to deal with sick employees have a high cost-benefit factor due to reduced sick leave [45].

A program for managers focussing on work-stress associated psychosomatic health issues and the interaction with affected employees was developed by Schwarz and colleagues [46,47]. It combines theoretical input containing facts about mental illness and work stress models with a focus on ERI with interactive group work on personal stressors and case discussions on dealing with affected employees. They successfully demonstrated that the program was effective long-term at reducing knowledge-based stigma among managers from a German industrial company, with the limitation that no control group was investigated. We aimed at further validating the findings by the inclusion of a randomized control group. We expected to find a change in leaders’ attitudes and stigma related to mental health, and in addition we wanted to explore whether intervention improves stress-related variables or the mental health of managers. 

## 2. Materials and Methods

### 2.1. Sample

The study took place in cooperation with a metalworking company and its health insurance fund. The sample description can be found in Section 3.1 Descriptives. The company’s managers were informed about the project by email in November 2018. Being a manager at the cooperating company was the only prerequisite for participating in this project. Managers who showed an interest in the study were informed verbally and in writing about the study details and gave their written consent to participate. A total of 100 managers registered by February 2019. Subsequently, the Institute for Epidemiology and Biometry of the University of Ulm randomly assigned the participants to either the intervention or waiting-list control group. Stratified randomization was performed based on worksite and position in the company. Managers randomized to the intervention group received sign-in dates for the training. 

The study protocol was approved by the Ethics Committee of Ulm University (326/16) and the study was conducted in accordance with the Declaration of Helsinki. 

### 2.2. Training

The content of the one-day workshop was based on the program developed by Boysen and colleagues [46]. The training lasted about 7.5 h and took place at two different company locations with a maximum of 15 and a minimum of five participants per date. The workshop started with input about mental and somatic stress-related diseases and possible connections to workplace factors. Common work stress models including the ERI model were discussed.

This was followed by interactive group work on personal stressors. Before the group discussion started, the managers first worked alone and then in pairs on a worksheet identifying early signs of stress in a personal situation. Based on cognitive-behavioral therapy, they recorded their thoughts, feelings, physical reactions, and behavior. In addition, the managers were asked to identify possible solutions with which they had previously succeeded in improving the situation described. Afterwards, the records were discussed in the whole group and the session was rounded off with a summary of ways to increase resilience. Working on personal stress experience should on the one hand increase the self-efficacy of the managers in stressful situations and thus lower stress levels. Raising awareness of the fact that everyone develops psychological and physical symptoms under stress, on the other hand, should help develop a more compassionate attitude towards stressed employees.

The last session before lunch was a presentation on how to communicate with sick or burdened employees. After lunch, the topic was examined in more detail along with a discussion of cases from the managers’ experience. The managers were asked to share a recent experience with a mentally unwell employee which they felt was difficult to handle. Two or three cases were discussed in detail. The discussion was moderated on a psychodynamic basis with a focus on relationship building and accompanying emotions. If the manager agreed, a role-play of a short situation between the manager and the employee was conducted, the thoughts and feelings of the players and the viewers were discussed and the situation was replayed. The goal of this part was to raise understanding of the behavior of the employee and to increase the self-efficacy and confidence of the manager in contact with the respective employee.

Afterward, the company physician described existing support services at the site, followed by a short presentation on external support services such as psychotherapy within the statutory healthcare system. The workshop ended with a summary and feedback. The workshops were each led by a specialist in psychosomatic medicine and psychotherapy, accompanied by another physician so that there were two trainers present per workshop. 

### 2.3. Questionnaires

Data were collected with paper-pencil questionnaires at baseline and after three months. Socio-demographic variables such as age, gender, hierarchical position in the company, level of education, number of children, and worksite were only assessed at baseline. 

To capture cognitive-based mental health stigma and attitudes, the Mental Health Knowledge Schedule (MAKS, [48]) and the Social Distance Scale [49] were used. In its original form, the MAKS uses six items (e.g., “Medication can be an effective treatment for people with mental health problems”) to inquire about cognitive-based stigma regarding mental illnesses [48]. For the evaluation of the training, only the first four items were used. Items number five and six were dropped due to concerns about their content validity. The statement for item five (“People with severe mental health problems can fully recover”) is not necessarily true for every mental illness, e.g., there are chronic courses of schizophrenia without total remission [50]. The sixth item (“Most people with mental health problems go to a health care professional to get help”) relates to the supply situation concerning the utilization of medical services. Unfortunately, a large proportion of mentally ill people in Germany still do not seek professional help [51]. Therefore, a rejection of this item may reflect not stigma but knowledge. The scale of the items ranged from one “completely agree” to five “not at all”. In addition, respondents could select “I do not know” as an answer. In the present analysis, these answers were counted as missing. The mean value of the scale was calculated, with higher values indicating more stigma [48]. Cronbach’s α for this scale was α = 0.65. 

The Social Distance Scale is a measurement tool for attitudes towards certain groups. The wish to avoid contact with the group concerned is considered a proxy for stigma [52]. For this study, we used the five items (e.g., “How willing would you be to make friends with the person?”) from Link and colleagues [49] and described the target for the questions as mentally ill but in treatment. The Social Distance Scale ranges from zero “definitely” to four “definitely not” with higher values indicating a stronger aversion to proximity. 

In addition to a change in stigma, we wanted to explore whether the short session on dealing with personal stressors and the better preparedness for dealing with affected employees could change the managers’ experience of work stress. Therefore, the following questionnaires were also collected.

For measuring stress in general, the validated four-item short form of the perceived stress scale was used (PSS-4), [53,54]. The scale for the items (e.g., “How often did you feel confident in dealing with your tasks and problems?”) ranged from 0 to 4. The values of the four items were combined into a sum score, with higher scores representing higher stress levels. The internal consistency of the scale is sufficiently high at α = 0.77 [53].

As work-related questionnaires, we used the German versions of the well-validated Effort-Reward Imbalance Questionnaire (ERI) [55] and the Irritation Scale (IS-8) [56]. The ERI measures work-related effort and reward with two separate scales. The effort scale consisted of six items (“I have a lot of responsibility in my job”) with a range of one to five. Higher values mean that more effort is invested in daily work-life. To measure subjective reward as a result of invested effort, eleven items (“I experience adequate support in difficult situations”) were used (range: 11–55). Financial reward, job security, opportunities for advancement, and esteem were considered rewards in the questionnaire [57]. The reliability of the scale was sufficiently high with a Cronbach’s α between α = 0.62 and α = 0.90.

The irritation scale captures subjective cognitive and emotional strain concerning work (e.g., “Even at home I often think of my problems at work”). It has been successfully validated for different industries [58]. The global score is the sum of all items (range 1–7), with higher scores representing higher levels of irritation. The reliability of the scale ranged from a Cronbach’s α of 0.85 to 0.93. 

Symptoms of depression and anxiety in the last two weeks were assessed using the four-item short form of the Patient Health Questionnaire (PHQ-4), [59]. Two items each ask about symptoms of anxiety and depression, ranging from zero “not at all” to three “nearly every day”. Higher scores indicate greater symptom burden. Past research has demonstrated that the PHQ-4 is a valid and time-efficient screening instrument for depression and anxiety [59,60]. Reliability in the validation studies was α = 0.82 and α = 0.85, respectively.

### 2.4. Data Sample

Out of one hundred included participants, 19 did not return the first questionnaire and 21 did not return the second questionnaire after training, resulting in *n* = 81 for the first and *n* = 79 for the second questionnaire. It occurred at both measurement points that some participants did not complete all of the questionnaires presented. This resulted in slight differences in the analyzed samples for the individual scales. As some participants also did not submit a questionnaire for time point 1 but did so for time point 2, the complete analysis samples for hypothesis testing ranged from *n* = 67 to *n* = 70 (see Figure 1). 

### 2.5. Design & Data Analysis

The study was conducted as a 2 × 2 mixed design (factor 1 (group): intervention or control group; factor 2 (time): baseline, three months later). The MAKS, ERI, PHQ-4, PSS, SoDi, and IS-8 were collected as dependent variables. 

Preliminary statistical (Shapiro–Wilk) and graphical analyses revealed that the data was not distributed normally. Therefore, non-parametric tests were applied for hypothesis testing. To compare the intervention and control groups at baseline, the t-test for independent samples was used for interval data and the chi-square test was used for nominal data. No statistically significant differences between intervention and control group for age, gender, hierarchical position in the company, level of education, number of children, or worksite were found at baseline.

To test the hypotheses, differences in the questionnaire scores were calculated by subtracting the baseline values from the post-intervention values. The new variables were then tested for group differences using the Mann–Whitney–U test. All data management and statistics were conducted using SPSS Statistics for Windows, version 25 (SPSS Inc., Chicago, IL, USA). A *p*-value smaller than 0.05 (two-sided) was considered statistically significant.

## 3. Results

### 3.1. Descriptives

The socio-demographic variables for all participants who completed at least one questionnaire are shown in Table 1. The participants were predominantly men averaging 48.07 (SD = 8.65) years old. Among the participating managers, positions ranged from shift supervisor to vice-president. 

### 3.2. Quantitative Results

The median, interquartile range, and results of the Mann–Whitney–U test for each measurement point for both groups are shown in Table 2. A significant difference was found between the groups according to the MAKS questionnaire (U = 417.00, *r* = 0.27, *p* = 0.040) and the IS (U = 371.50, *r* = 0.31, *p* = 0.011). The MAKS median values slightly decreased in the IG accompanied by a slight increase in the WG. The Irritation Scale showed lower irritation in the IG at baseline, while values did not differ between IG and WG at follow-up. A graphical representation of the significant results is shown in Figure 2. There were no significant changes in the Effort–Reward Imbalance (ERI), the Perceived Stress (PSS), or the Social Distance (SoDi) scale, nor regarding depression and anxiety symptoms (PHQ-4). 

## 4. Discussion

The present study showed changes in knowledge-based stigma in the intervention group measured using MAKS three months after training. This is in line with the findings published by Schwarz and colleagues [47]. 

The change in MAKS can be attributed to the training content. Participants were given a theoretical insight into the interplay between work stress and mental illnesses and discovered personal stress symptoms. Additionally, examples of how to deal with stressed employees were discussed in the context of collaborative casework. These lessons were designed to increase cognitive understanding and sensitivity regarding mental health problems. Through this improved understanding of mental illness, participants should have gained a more realistic view of mental illness. This in turn makes it plausible that cognitive stigma, which was based on unrealistic assumptions, had been reduced. Similar effects have been demonstrated in the past after anti-stigma training for medical students and police officers [40,61].

In contrast, there was no change in the more affective-based SoDi. On the one hand, managers might need to be more involved on a personal level to achieve changes in this area. On the other hand, the values obtained by the SoDi were near to the minimum, corresponding to a floor effect without the relevant possibility for improvement. The values corresponded to those found by Schwarz and colleagues [47]. As with their study, a possible rationale lies in social desirability, furthered by the social working culture of the company. In addition, participation in the training was voluntary, so it can be assumed that there was self-selection of managers who were generally more open to the topic of mental health and who therefore generally had a lower tendency to stigmatize. 

There was no improvement in subjectively perceived emotional and cognitive stress revealed by the work context questionnaires or the Perceived Stress Scale. Regarding the baseline of the Irritation Scale, which represents one aspect of emotional stress, participants of the IG showed lower values than the waiting control group (*MDN*(*IG*) = 17.50; *MDN*(*WG*) = 22.00) which converged to those of the waiting group at follow-up. This contrasts with other studies of stress management training, which are often associated with improvement in the experience of stress inside and outside the work context [62,63,64]. However, comparison of the norm values of the PSS and Irritation Scale shows that the values of the intervention group were comparatively low at baseline [53,58]. It has been shown in other research that assignment to the intervention group can already lead to a change in participants’ attitudes [65]. It might be possible that in this case, the group allocation had already affected the intervention group, but the effect diminished over three months and the values returned to the norm, while the intervention was not sufficient to slow down this development. Improved interaction with employees did not lead to significant stress reduction for managers. Dealing with sick employees is just one part of managers’ daily stress, and this part might be too small to be represented as a measurable change in overall stress. Likewise, the session on personal stressors, which was supposed to lead to a more compassionate perspective through self-experience and to improve dealing with stressors, was not sufficient to cause an improvement. If this is the intended focus, other approaches that concentrate more directly on the stress management of the leaders are more promising [62].

The main focus of this training was to reduce stigma and to improve dealings with stressed employees. However, possible positive effects on employees’ experiences of stress were not evaluated. In future investigations with similar training content, it would be beneficial to include measurement instruments for this purpose, e.g., by surveying the manager’s assigned team members regarding a change in the manager’s behavior. 

Despite considerable research on stigma, it remains a major social problem. Especially in the workplace, people with mental illness are at risk of prejudice and social ostracism. Managers have an important function as a frequent point of contact. The results of this naturalistic study demonstrated that it is possible to reduce cognitive-based stigma towards people with mental illness with one day of training. The advantage of training compared to public campaigns is that it can be used in a very targeted way. It is reasonable to assume that the effectiveness of training increases if the target population has higher baseline stigma levels. Therefore, such training could be implemented by companies as a result of needs analysis concerning their employed managers. 

In summary, this small but randomized controlled study, implemented in a naturalistic occupational setting, was able to further validate the findings that one-day training is sufficient to reduce cognitive stigma. At the same time, there was no reduction observed for managers’ stress levels. Future investigations should also focus on this. Additionally, booster sessions could be added after the training to consolidate the achieved changes. 

## 5. Limitations

One limitation of the present work is reduced generalizability due to the sample composition. The participants were all from the same company and had a high level of education. Thus, no statements can be made about other industry sectors or small to medium enterprises. At the same time, managers were mainly male, so the findings cannot be broadly generalized to other genders. Furthermore, the offer was voluntary, so no statement can be made about the effectiveness of the workshop for managers who show little interest in this topic. Though we used a randomized controlled design, we had a high drop-out rate which could compromise the design. Still, we found no differences between participants and dropouts. 

## 6. Conclusions

The presented one-day intervention was sufficient to reduce cognitive stigma after three months. No effect was found for stress-related variables or affective stigma, so a different approach should be pursued in this regard. Generally, future research on a more diverse population regarding gender, company size, and industry sector is needed to further validate the observed effects.

## Figures and Tables

**Figure 1 ijerph-19-04139-f001:**
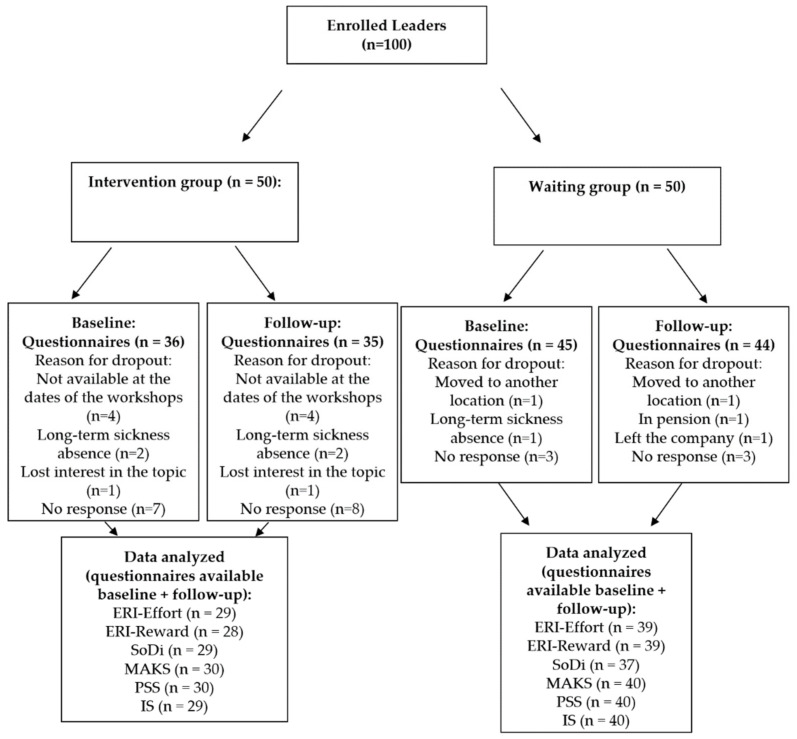
Recruitment Flowchart.

**Figure 2 ijerph-19-04139-f002:**
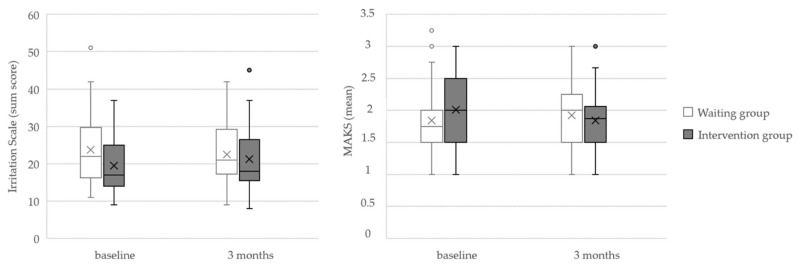
Boxplots for the Mental Health Knowledge Scale (range: 1–5) and Irritation Scale (range: 1–64).

**Table 1 ijerph-19-04139-t001:** Characteristics of the study population.

	Intervention Group (*n* = 30)	Control Group (*n* = 40)	*p*
Age *M* (SD)	48.17 (8.17)	47.68 (9.25)	0.818
Male gender *n* (%)	29 (96.7%)	37 (92.5%)	0.457
Underage children *M* (SD)	0.97 (1.07)	1.24 (1.02)	0.286
High school diploma or comparable%	60.0%	57.5.%	0.834
Company site	14/16	21/18	0.554
Position ^1^			0.902
A *n* (%)	5 (16.7%)	6 (15.0%)	ns.
B *n* (%)	13 (43.3%)	19 (47.5%)	ns.
C *n* (%)	5 (16.7%)	7 (17.5%)	ns.
D *n* (%)	7 (23.3%)	8 (20.0%)	ns.

^1^ Position in the company ranging from stage A (lowest) to stage D (highest). A = Chief Officer, Vice-President; B = Director, Head of, Production Manager; C = Shift Supervisor, Bachelor Professional; D = Team Leader, HR Business Partner.

**Table 2 ijerph-19-04139-t002:** Medians, interquartile range, Mann–Whitney–U test statistics, and *p*-values for all variables collected.

	*MDN*_T1_ (*IQR*)		*MDN*_T2_ (*IQR*)		U	*p*
	IG	WG	IG	WG		
ERI-Effort ^1^	15.00 (14.00;18.00)	16.00 (14.25;19.00)	17.00 (14.00;20.00)	16.00 (14.00;19.75)	459.00	0.183
ERI-Reward ^2^	48.00 (39.00;43.00)	50.00 (39.50;54.00)	45.00 (33.75;53.00)	51.00 (40.00;53.00)	478.50	0.389
PHQ-4 ^3^	1.00 (0.00;2.00)	1.50 (1.00;3.00)	1.00 (0.00;3.00)	1.50 (0.25;3.00)	545.00	0.382
SoDi ^4^	0.80 (0.40;1.55)	0.80 (0.20;1.50)	0.80 (0.40;1.20)	0.80 (0.40;1.55)	524.50	0.365
MAKS ^5^	2.00 (1.50;2.31)	1.75 (1.50; 1.75)	1.75 (1.50;2.25)	2.00 (1.50;2.25)	417.00	0.040 *
PSS ^6^	3.00 (2.00;5.00)	4.00 (3.00;6.00)	4.00 (2.00;6.00)	4.00 (3.00;6.00)	439.00	0.051
IS-8 ^7^	17.50 (14.00;24.00)	22.00 (16.50;29.50)	20.00 (16.00;28.50)	21.00 (17.25;30.00)	371.50	0.011 *

^1^ Effort–Reward Effort Subscale (range: 6–30), ^2^ Effort–Reward Reward Subscale (range: 11–55), ^3^ Patient Health Questionnaire-4 (range: 0–12), ^4^ Social Distance Scale (range: 1–4), ^5^ Mental Health Knowledge Scale (range: 1–5), ^6^ Perceived Stress Scale (range: 0–20), ^7^ Irritation Scale (range: 1–64). * Significant at the 0.05 level.

## Data Availability

The data that support the findings of this study are available on request from the corresponding author.

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
