# Peer review of "A Single-Day Training for Managers Reduces Cognitive Stigma Regarding Mental Health Problems: A Randomized Trial"

_ijerph, 2022, doi:10.3390/ijerph19074139_

Round 1

Reviewer 1 Report

The paper assesses the impact of a theoretical training, group work therapy, and group discussion on improving mental health and irritation scale and reducing stress. The study is longitudinal with three questionnaires administered at baseline and three months after. The following are my comments:

  • The topic by itself is not very new and the researchers have focused on awareness about mental health issues as an intervention by itself for reducing stress. The literature review briefly mentions this fact but additional attention and details would really benefit the need for this research.
  • The literature review section is missing. The study needs justification as to why it was conducted and summarize recent research in the domain.
  • The training module includes an awareness session, a group discussion to aid awareness again, and finally, a session that discussed how to communicate with stressed colleagues. The interventions need further justification to show how they could help participants learn to manage their personal stress.
  • The social stigma associated with mental illness and how it affects work should be emphasized more in the literature review.
  • As the results of the study have indicated no significant impact on Effort-Reward Imbalance, the Perceived Stress, or the Social Distance scale or depression and anxiety symptoms, the researchers should consider focusing on how awareness campaigns can help reduce irritation and build empathy.
  • Lines 277-278: “Therefore, it is plausible that the training reduced cognitive stigma in the participants”. This statements needs clarification.
  • One of the key reasons why there was not much impact seen in behaviour of the respondents is aptly pointed out by the researchers. “Dealing with one's own stressful experience took up only a small portion of the training time”. It is recommended that the research objectives are revised to pinpoint a better fit between what was intended to be achieved and the interventions used.
  • A revision of objectives would also help strengthen the focus of the study: remove stigma or making respondents more self-aware about mental illness?

Reviewer 2 Report

This research examined the effectiveness of a training program to promote the managers’ mental health. This seems to be interesting and useful for workplace health. However, there are several concerns on the paper writing.

(1) The research gap is not very clear in the Introduction section. They should focus directly on the mental health of managers rather than the workforce in general. Why training program for managers is more important? I suggest that the Introduction can be rewritten much clearly.  

(2) There is lack of detailed discussion on health training intervention, especially the current literature. Why training intervention rather than other interventions should be selected? They mentioned “Prevention approaches which aim at reducing stress and thereby mental illnesses at the workplace are commonly based on the above mentioned models” in Page 2 Line 56. I merely found the ERI model here.

(3) In Materials and Methods, they used several scales such as ERI, SoDi, MAKS, PSS and IS in the research. Considering that many of them were not significant at the end, please tell the readers why all these scales were selected and what is the relationship between them much clearly.  

(4) They mentioned “To test the hypothesis” in Page 5 line 214. What is the hypothesis?

(5) The “six months later” should be “three months later” in Page 5 line 205

(6) The theoretical and practical implications should be emphasized in Discussion

Round 2

Reviewer 2 Report

Thanks for adopting my suggestions.The authors have responded all comments I gave in the first round review process